# Ocular movement abnormalities and ptosis after glaucoma surgery: A retrospective decade long analysis

Carlo Catti[1,2,3], Federica Tessitore[1,2], Daniele Ferrari[1,2], Federica Milanesi[1,2], Silvia Acerra[1,2], Luigi Peci[1,2], Arianna Rizzi[1,2], Silvio Semeria[1,2], Irene Schiavetti[4], Carlo Alberto Cutolo[1,2], Michele Iester[1,2], Aldo Vagge[1,2]*

1 IRCCS Ospedale Policlinico San Martino, University Eye Clinic of Genoa, Genoa, Italy, 2 Dipartimento di Neuroscienze, Riabilitazione, Oftalmologia, Genetica e Scienze Materno-infantili (DINOGMI), Università di Genova, Genoa, Italy, 3 Dipartimento di Medicina Sperimentale (DIMES), Università di Genova, Genoa, Italy, 4 Dipartimento di Scienze della Salute (DISSAL), Università di Genova, Genoa, Italy

* aldo.vagge@unige.it

## Abstract

### Background/Aims

To evaluate the prevalence of ptosis, strabismus and the combination of both after glaucoma surgery and determine which kind of surgery is most likely to be linked to these complications.

### Methods

A total of 705 clinical records of patients who underwent glaucoma surgery at the University Eye Clinic of IRCCS Ospedale Policlinico San Martino, Genoa, from January 1, 2010, to December 31, 2020, were retrospectively evaluated. Surgery procedures were divided in three groups: "Ab interno", "Ab externo" and "Muscle isolation".

### Results

Out of all patients 26 developed ptosis alone (3.7%), two developed bilateral ptosis, with a mean of 2.2 ± 1.08 procedures per patient. Highest incidence of ptosis was noticed in patients who underwent muscle isolation surgery (5.7%). Twelve patients developed strabismus alone (1,7%), four underwent unilateral surgery and eight bilateral surgery, with an average of 3.3 ± 1.78 procedures per patient. Strabismus was more frequent following extraocular muscle manipulation surgery (7.5%), showing a statistically significant difference (OR: 6.57; 95% CI: 1.71–21.65; p = 0.003) Seven patients developed both strabismus and ptosis (1.0%), one patient with bilateral ptosis. Four underwent bilateral surgery and three underwent unilateral surgery. The mean number of surgeries was 2.9 ± 1.07, showing a statistically significant difference compared to the uncomplicated glaucoma group (OR: 1.58; 95% CI: 1.01–2.25;

**Editor:** Nader Hussien
Lotfy Bayoumi, Alexandria University Faculty of Medicine, EGYPT

**Data availability statement:** All relevant data are within the paper and its Supporting information files.

**Funding:** The author(s) received no specific funding for this work.

**Competing interests:** The authors have declared that no competing interests exist.

p = 0.02). The prevalence of both ptosis and strabismus was higher after muscle isolation surgery (1.9%).

## Conclusion

Ptosis, strabismus and the combination of both are rare complications after glaucoma surgery, mostly linked to surgery with muscle isolation.

---

## Introduction

Glaucoma encompasses a broad spectrum of diseases characterized by progressive optic neuropathy, marked by morphological alterations in the optic nerve head and retinal nerve fiber layers due to ganglion cell death, and progressive visual field impairment often associated with elevated intraocular pressure (IOP).

Globally, glaucoma is the leading cause of irreversible vision loss and ranks second in Western countries after age-related macular degeneration. Approximately 76 million people worldwide were affected by glaucoma in 2020, with projections suggesting an increase to 112 million by 2040 [1,2].

Topical medical therapy remains the primary treatment modality; however, surgical interventions are pursued when adequate IOP reduction cannot be achieved through pharmacological measures alone. The first documented surgical technique, trabeculectomy, was introduced by Cairns [3]. Modern glaucoma surgery includes a variety of implantable devices, both valved and non-valved, which are minimally invasive and effective. These devices have gained popularity among glaucoma surgeons due to their ability to maintain stable IOP levels, minimal invasiveness, and reduced incidence of post-procedural complications [4–6].

Despite recent advancements in surgical techniques, several complications remain, including choroidal effusion, suprachoroidal hemorrhage, vitreous hemorrhage, endophthalmitis, hypotony or hypertony, cataract, hyphema, shallow anterior chamber, corneal edema, bleb leakage, strabismus, and ptosis [7–9].

Strabismus and ptosis, in particular, may arise⁵ due to anatomical alterations in the eye or the extraocular muscles responsible for ocular motility [10,11].

Strabismus is a neuromuscular condition in which the visual axes of the eyes are misaligned relative to the object of fixation,resulting in improper binocular fusion of images and symptoms such as diplopia or visual confusion. The onset of strabismus is often facilitated by the presence of pre-existing conditions such as amblyopia and reduced visual acuity, the latter being often a consequence of advanced glaucoma [12]. Ptosis, characterized by drooping of the upper eyelid, may obscure the pupil and interfere with the visual field, often presenting aesthetic concerns for patients as well [13].

To date, there are no large-scale studies focusing specifically on strabismus and ptosis following glaucoma surgery, nor do existing studies include a substantial patient cohort with a diversity of surgical techniques [7–9].

The objective of this study is to assess the incidence of strabismus, diplopia, and ptosis following glaucoma surgery.

## Methods

This retrospective observational study, conducted in compliance with the tenets of the Declaration of Helsinki, received approval from the IRCCS Policlinico San Martino Ethics Committee, Genoa. All clinical records of patients who underwent glaucoma surgery at the University Eye Clinic of IRCCS Ospedale Policlinico San Martino, Genoa, from January 1, 2010, to December 31, 2020, were examined between March 1 and April 30, 2024 and included. Medical records, including patient history, comprehensive ophthalmologic evaluations, and surgical reports, were thoroughly reviewed. For each patient written consent for surgery and research purpose acquired and data analysed anonymously.

The study encompassed various types of filtering surgeries, such as trabeculectomy, CO2 Laser-Assisted Sclerectomy Surgery (CLASS), Xen implant, PreserFlo microshunt, Baerveldt implant, ExPress shunt, and cyclocryotherapy. Patients were categorized into four groups according to postoperative outcomes:

1) "Uncomplicated Glaucoma" Group: Patients with no subjective or objective indications of strabismus or ptosis during any eye examinations.

2) "Ptosis" Group: Patients who developed ptosis following surgery.

3) "Strabismus" Group: Patients presenting with strabismus or binocular diplopia after surgery.

4) "Ptosis with Strabismus" Group: Patients positive for both ptosis and strabismus following surgery.

Patients were further categorized based on the type of filtering procedure performed, organized into three main surgical categories:

1) Surgery without Conjunctival Opening – Ab Interno: Procedures such as Xen implant and cyclocryotherapy. The latter being a noninvasive procedure that does not require conjunctival opening, has been included in this classification for the recorded effects on ocular motility and ptosis.

2) Surgery with Conjunctival Opening – Ab Externo: Including PreserFlo implant, trabeculectomy, CLASS deep sclerotomy, ExPress implant, and revision of filtering surgeries.

3) Surgery with isolation and extraocular muscle manipulation: Specifically, the Baerveldt implant.

These procedures are listed in order from least to most invasive. Each category included patients who underwent the same type of procedure, including those who had multiple surgeries or surgical revisions. In cases involving multiple surgeries on the same eye, the eye was categorized under the most invasive procedure. Patients with bilateral surgeries were classified according to the most invasive procedure performed.

All patients underwent peribulbar anesthesia before surgery. Each eye was covered for each surgery with a sterile drape and underwent accurate disinfection before and after the sterile drape application. A standardized wheel speculum was used during surgery to keep the eye open. All surgeries were performed by an expert surgeon and lasted lesser than one hour. If needed, a translimbal corneal traction suture was used to expose the sclera.

Exclusion Criteria: Patients were excluded if they had glaucoma surgery at our institution but follow-up was conducted elsewhere, if medical records were incomplete, if they had monocular diplopia, strabismus, or ptosis prior to glaucoma surgery, or if they were lost to follow-up less than one year after filtering surgery.

Demographic data (age and sex), subjective sensorimotor symptoms (e.g., diplopia, abnormal motility, and ptosis), visual acuity, and ocular alignment in the five primary gaze positions were documented from medical records. Each patient with strabismus, diplopia, or ptosis was examined by both a pediatric ophthalmologist experienced in strabismus evaluation and surgery and an oculoplastic surgeon. The nine cardinal positions were evaluated and cover-uncover and alternate cover tests were performed in all gaze positionsat both near and far distances. For patients with visual acuity worse than 20/200 or unable to maintain fixation, the Krimsky test was used to assess ocular alignment.

Postoperative ptosis was assessed by measuring Marginal Reflex Distances 1 and 2 (MRD1 and MRD2), along with evaluation of levator palpebrae superioris function, performed by an experienced oculoplastic surgeon. Ptosis severity was classified based on MRD1 values as mild (2–3 mm), moderate (1–2 mm), or severe (0–1 mm). Descriptive analysis is reported as mean with standard deviation for continuous variables and as absolute and relative frequency for categorical variables. The analysis of factors associated with three outcomes (ptosis, strabismus, and the combination of the two outcomes) was performed using a univariate logistic regression model, followed by a multivariate analysis if more than one factor was significant in the univariate analysis. A p-value of less than or equal to 0.05 was considered significant. The analysis was performed in R.

## Results

The study included 705 medical records of patients who underwent glaucoma surgery. Among these, 361 were male (51.2%) and 344 were female (48.8%), with a mean age of $67.3 \pm 15.75$ years. A total of 654 patients were excluded due to not meeting the inclusion criteria previously described.

Of the included patients, 473 (67.1%) underwent unilateral surgery, and 232 (32.9%) had bilateral surgeries, with an average of $1.8 \pm 1.12$ surgeries per patient. Patients who developed strabismus, diplopia, or ptosis were compared with those in the uncomplicated glaucoma group who underwent the same type of surgery.

The uncomplicated glaucoma group comprised 660 patients (93.6%): 334 males (50.6%) and 326 females (49.4%), with a mean age of $67.7 \pm 15.57$ years.

The ptosis group (Table 1) included 26 patients (3.7%), of whom 16 were male (60.5%) and 10 were female (38.5%). Two of these patients exhibited bilateral ptosis.

The mean age in this group was $62.8 \pm 19.24$ years. There were no statistically significant differences in age or sex distribution between the ptosis group and the uncomplicated group.

The strabismus group (Table 2) consisted of 12 patients (1.7%), with 7 males (58.3%) and 5 females (41.7%). The mean age of this group was $58.3 \pm 17.54$ years. While no statistically significant difference was observed in sex distribution, patients who developed strabismus following glaucoma surgery were found to be younger than those with uncomplicated outcomes ($p = 0.04$).

**Table 1. Characteristics by ptosis.**

| | | *Absent* | *Present* |
|---|---|---|---|
| | | *(N = 679)* | *(N = 26)* |
| *Age* | | *67.5 ± 15.59* | *62.8 ± 19.24* |
| Sex (male vs female) | Females | 334 (97.1%) | 10 (2.9%) |
| | Males | 345 (95.6%) | 16 (4.4%) |
| Side eye (bilateral vs monolateral) | Monolateral | 453 (95.8%) | 20 (4.2%) |
| | Bilateral | 226 (97.4%) | 6 (2.6%) |
| Number of interventions | | 1.8 ± 1.13 | 2.2 ± 1.08 |
| Ab_internal surgery | Not executed | 451 (95.8%) | 20 (4.2%) |
| | Executed | 228 (97.4%) | 6 (2.6%) |
| Ab_external surgery | Not executed | 154 (97.5%) | 4 (2.5%) |
| | Executed | 525 (96.0%) | 22 (4.0%) |
| Isolation | No | 629 (96.5%) | 23 (3.5%) |
| | Yes | 50 (94.3%) | 3 (5.7%) |

**Table 2. Characteristics by strabismus.**

| | | *Absent* | *Present* |
|---|---|---|---|
| | | *(N = 693)* | *(N = 12)* |
| *Age* | | 67.5 ± 15.69 | 58.3 ± 17.54 |
| *Sex (male vs female)* | *Females* | 337 (98.0%) | 7 (2.0%) |
| | *Males* | 356 (98.6%) | 5 (1.4%) |
| *Side eye (bilateral vs monolateral)* | *Monolateral* | 469 (99.2%) | 4 (0.8%) |
| | *Bilateral* | 224 (96.6%) | 8 (3.4%) |
| *Number of interventions* | | 1.8 ± 1.09 | 3.3 ± 1.78 |
| *Ab_internal surgery* | *Not executed* | 464 (98.5%) | 7 (1.5%) |
| | *Executed* | 229 (97.9%) | 5 (2.1%) |
| *Ab_external surgery* | *Not executed* | 156 (98.7%) | 2 (1.3%) |
| | *Executed* | 537 (98.2%) | 10 (1.8%) |
| *Isolation* | *No* | 644 (98.8%) | 8 (1.2%) |
| | *Yes* | 49 (92.5%) | 4 (7.5%) |

Seven patients (1.0%), (Table 3) developed both strabismus and ptosis including six males (85.7%) and one female (14.3%); of these, one patient exhibited bilateral ptosis. This group had a mean age of 67.7 ± 11.16 years, with no statistically significant differences in age or sex distribution compared to the uncomplicated glaucoma group.

### Ptosis group

In the ptosis group, 20 patients underwent unilateral surgery and 6 had bilateral surgery; of these, 2 patients developed bilateral ptosis, with a mean of 2.2 ± 1.08 procedures per patient. 15 of them, including the two cases of bilateral ptosis, developed a mild ptosis (57,7%), 9 a moderate ptosis (34,6%) and 2 of them (7,7%) a severe ptosis.

The specific surgical procedures associated with ptosis are listed in Table 4, with trabeculectomy plus revision being the most frequently associated, followed by trabeculectomy alone.

When categorizing patients by the three main types of surgery (Table 1), the incidence of ptosis was distributed as follows: 3 cases with ab interno surgery, 20 cases with ab externo surgery, and 3 cases with extraocular muscle isolation.

**Table 3. Characteristics by ptosis and strabismus.**

| | | *Absent* | *Present* |
|---|---|---|---|
| | | *(N = 698)* | *(N = 7)* |
| *Age* | | 67.3 ± 15.80 | 67.7 ± 11.16 |
| *Sex (male vs female)* | *Females* | 343 (99.7%) | 1 (0.3%) |
| | *Males* | 355 (98.3%) | 6 (1.7%) |
| *Side eye (bilateral vs monolateral)* | *Monolateral* | 470 (99.4%) | 3 (0.6%) |
| | *Bilateral* | 228 (98.3%) | 4 (1.7%) |
| *Number of interventions* | | 1.8 ± 1.12 | 2.9 ± 1.07 |
| *Ab_internal surgery* | *Not executed* | 468 (99.4%) | 3 (0.6%) |
| | *Executed* | 230 (98.3%) | 4 (1.7%) |
| *Ab_external surgery* | *Not executed* | 157 (99.4%) | 1 (0.6%) |
| | *Executed* | 541 (98.9%) | 6 (1.1%) |
| *Isolation* | *No* | 646 (99.1%) | 6 (0.9%) |
| | *Yes* | 52 (98.1%) | 1 (1.9%) |

**Table 4. Number of positive ptosis and/or strabismus cases analysing each type of surgery or combination of surgeries separately.**

| Cathegory | Surgery | Ptosis | Strabismus | Ptosis+Strabismus |
|---|---|---|---|---|
| Ab interno | Xen | 3 | 1 | 1 |
| Ab externo | Trabeculectomy | 6 | 2 | – |
| | Trabeculectomy+revisions | 9 | 1 | 2 |
| | Xen+Trabeculectomy | – | 3 | – |
| | Xen+revisions | 3 | – | – |
| | Ex-press | – | 1 | – |
| | Ex-press+revisions | 2 | – | – |
| | Ex-press+Cyclocryoablation | – | – | 2 |
| | CLASS | – | – | 1 |
| Muscle isolation | Baerveldt | 1 | 2 | – |
| | Baerveldt+revisions | 1 | – | – |
| | Trabeculectomy+Bearveldt | 1 | – | – |
| | Ex-press+Baerveldt | – | 1 | – |
| | Trabeculectomy+ Baerveldt+ Cyclocryoablation | – | 1 | – |
| | Ex-press+Baerveldt+Cyclocryoablation | – | – | 1 |
| Total | | 26 | 12 | 7 |

The highest incidence of ptosis was observed in patients who underwent surgery involving muscle isolation (5.7%), followed by ab externo surgery (4.0%) and, lastly, ab interno surgery with the lowest incidence (2.6%).

No statistically significant differences were noted in terms of demographics, the number of surgeries, or the types of surgeries when compared to the uncomplicated glaucoma group.

## Strabismus group

Among the patients who developed strabismus, 4 underwent unilateral surgery and 8 bilateral surgery, with an average of 3.3±1.78 procedures per patient. Preliminary univariate analysis (Table 5) indicated that bilateral surgery, compared to unilateral surgery, significantly increased the risk of developing strabismus (OR: 4.19; 95% CI: 1.30–15.82; p=0.02).

Younger age was also associated with a significantly higher risk of developing strabismus (OR: 1.14; 95% CI: 1.06–1.29; p=0.04). Univariate analysis further revealed a statistically significant association between the total number of surgeries and the incidence of strabismus, with patients undergoing a higher number of interventions at increased risk (OR: 1.82; 95% CI: 1.34–2.42; p<0.001).

**Table 5. Results of univariate analysis.**

| Factor | Ptosis | Strabismus | Ptosis+strabismus |
|---|---|---|---|
| | OR(95%CI); p | OR(95%CI); p | OR(95%CI); p |
| Age | 0.98 (0.96–1.01); 0.13 | 1.14 (1.06–1.29); 0.04 | 1.00 (0.96–1.06); 0.95 |
| Sex (male vs female) | 1.55 (0.70–3.58); 0.29 | 0.68 (0.20–2.14); 0.51 | 5.80 (0.98–109.79); 0.11 |
| Side eye (bilateral vs monolateral) | 0.60 (0.22–1.43); 0.28 | 4.19 (1.30–15.82); 0.02 | 2.75 (0.60–14.05); 0.19 |
| Number of interventions | 1.12 (0.79–1.48); 0.48 | 1.82 (1.34–2.42);<0.001 | 1.58 (1.01–2.25); 0.02 |
| Ab internal surgery | 0.59 (0.21–1.42); 0.27 | 1.45 (0.42–4.58); 0.53 | 2.71 (0.59–13.87); 0.19 |
| Ab external surgery | 1.61 (0.61–5.58); 0.39 | 1.45 (0.38–9.51); 0.63 | 1.74 (0.29–33.01); 0.61 |
| Isolation | 1.64 (0.38–4.93); 0.43 | 6.57 (1.71–21.65); 0.003 | 2.07 (0.11–12.44); 0.50 |

Subsequent multivariate analysis, which included variables such as the total number of surgeries, laterality, and extraocular muscle manipulation, confirmed that only the number of previous interventions had a significant effect on the development of strabismus (OR: 1.47; 95% CI: 1.00–2.14; p = 0.048), while the effects of laterality and muscle isolation were no longer significant.

The distribution of strabismus cases by type of surgery is presented in Table 4. The procedures most commonly associated with strabismus included Xen implant combined with trabeculectomy, followed by single Baerveldt implantation and trabeculectomy alone (each associated with 2 cases). Strabismus was proportionally more frequent following extraocular muscle manipulation surgery (7.5%), showing a statistically significant difference from other surgical groups (OR: 6.57; 95% CI: 1.71–21.65; p = 0.003). The incidence of strabismus was 1.8% among patients treated with ab externo surgery and 2.1% among those treated

## Strabismus with ptosis group

A total of 7 patients developed both strabismus and ptosis, with 4 undergoing bilateral surgery and 3 undergoing unilateral surgery. Five of them (71,4%) developed a mild ptosis, two of them, including the patient with bilateral ptosis developed a moderate ptosis (28,6%) after glaucoma surgery. The mean number of surgeries in this group was 2.9 ± 1.07, showing a statistically significant difference compared to the uncomplicated glaucoma group (OR: 1.58; 95% CI: 1.01–2.25; p = 0.02) in univariate analysis.

The surgical procedures most frequently associated with the occurrence of both strabismus and ptosis are presented in Table 4. Notably, 2 cases followed trabeculectomy with revisions, and 2 cases followed Ex-Press implantation combined with cyclocryoablation. Additionally, 1 patient developed strabismus with bilateral ptosis; this patient had undergone Ex-Press implant surgery in the fellow eye. The incidence of both ptosis and strabismus was proportionally highest in cases involving extraocular muscle manipulation (1.9%), followed by ab interno surgeries (1.7%) and ab externo surgeries (1.1%) (Table 3).

Considering both the ptosis only and the ptosis+strabismus group, among all patients affected by ptosis (n = 33), 21.2% of them also developed strabismus (n = 7), highlighting a notable overlap between these conditions. Conversely, when examining all patients with strabismus as the sum of the strabismus only and the ptosis+strabismus group (n = 19), the association is even more pronounced: 36,8% of these patients (n = 7) also presented with ptosis.

Of the 19 patients diagnosed with strabismus, 5 reported experiencing diplopia (26.3%), while 14 did not report diplopia (73.7%). It is worth noting that 12 of these 19 strabismus cases (63.1%) occurred in patients who underwent bilateral filtering surgery. All patients showed signs of restrictive strabismus.

Regarding the type of strabismus observed, 2 patients presented with esotropia (10.5%), 4 with exotropia (21.1%), 9 with hypertropia (47.4%), and 4 with a combination of exotropia and hypertropia (21.1%).

## Discussion

The etiology of strabismus following the implantation of glaucoma drainage devices is multifactorial and may result from various factors including the mass effect of the implant, implant-induced restriction, adherence of retro-orbital fat, intraoperative trauma to extraocular muscles, muscle scarring, or displacement of the muscle pathway. Extensive encapsulation around the implant can limit ocular movement in the direction of the implant, while scarring and contractures may pull the eye towards it [14–16].

The most common form of postoperative strabismus observed is restrictive strabismus [15]. The preferred implant position is typically in the superotemporal quadrant of the orbit, as this area offers more available space and reduces the likelihood of interference with the oblique muscles. Consequently, exotropia and/or hypertropia of the ipsilateral eye have the highest incidence. Additionally, inferior placement of drainage implants appears to be more likely to cause diplopia, particularly in reading positions, due to the altered visual alignment [17–20].

Different types of drainage implants vary in size and design, resulting in distinct interactions with surrounding tissues. Our data indicate that larger implant sizes are associated with a higher incidence of strabismus [12]. The Baerveldt implant, for instance, was initially associated with restrictive strabismus primarily due to its bulk. However, after redesigns introduced fenestrations to reduce plate bulk, newer Baerveldt models demonstrated a lower incidence of strabismus compared to the older, non-fenestrated models [16–21]. These fenestrations allow fibrous tissue growth through the plate, anchoring it to the bleb roof, which helps reduce the risk of motility disturbances.

Nevertheless, not all cases of diplopia can be attributed to a large bleb. In cases with a low-volume bleb, diplopia may arise from alternative mechanisms, such as the formation of fibrotic adhesions between the outer capsule and adjacent tissues [20].

Our findings regarding the prevalence of motility disorders align with previous large-sample studies. In the multicenter randomized Tube vs. Trabeculectomy Study, new motility disturbances were observed in 5% of the 101 patients who received a Baerveldt glaucoma implant, whereas no motility issues were detected in the trabeculectomy group at 1-year follow-up [22]. Similarly, Robbins et al. reported that among 732 patients treated with an Ahmed valve implant, 29 (4%) developed persistent strabismus beyond six months post-surgery [16].

Interestingly, the prevalence of strabismus increases exponentially with the number of surgeries undergone. We hypothesize that this may be due to an elevated tissue reactivity and a greater tendency for scar formation in patients undergoing multiple surgical procedures. Additionally, repeated surgeries and multiple anesthesia exposures may further increase the risk of postoperative diplopia, as reflected by the higher incidence of strabismus in patients undergoing bilateral surgeries [23]. Table 5. shows significant univariate analysis results for such risk factors associated with strabismus. This trend is also observed in ptosis incidence, which varies across surgical techniques, ranging from 1.64% to 12.50%, and increases in patients who undergo revision or bilateral surgeries.

Regarding ptosis, our study includes the largest sample size and the broadest array of surgical techniques considered to date. Consistent with previous case series, our findings indicate a higher incidence of ptosis in patients undergoing combined surgeries [24].

Age appears to be a relevant factor in the development of ptosis. Although the ptosis group in our study did not show a statistically significant age difference from the control group, it did have a lower mean age. Some studies suggest that younger patients may be more prone to postoperative ptosis—a somewhat counterintuitive but significant finding. Younger patients, who have yet to experience age-related degenerative changes in the levator palpebrae muscle, may be more noticeably affected by surgery-induced ptosis than older patients, in whom age-related levator changes may already be present [24].

Other contributing factors to ptosis include prolonged use of antiglaucoma medications, which have been linked to periorbitopathy and ptosis through mechanisms such as Müller muscle degeneration and orbital adipocyte apoptosis [25], previous needling procedures, peribulbar anesthesia, multiple or lengthy surgeries, and even specific glaucoma types, for instance, pseudoexfoliation syndrome may increase ptosis risk due to cumulative tissue damage over time [26–28].

Patient age also plays a significant role in the development of strabismus. Younger patients, with a thicker Tenon's capsule and stronger fibrous bands between the lateral and superior rectus muscles, experience a higher incidence of reactive scarring [29]. Some studies, particularly in pediatric populations, advocate for supratenonian implantation of drainage valves to mitigate excessive fibrosis and reduce the risk of ocular motility impairment [30].

Additionally, younger patients, often with a shorter history of glaucoma, are more likely to notice diplopia due to better overall visual acuity and visual field preservation [16]. This group is inherently at a higher risk of developing strabismus, even before filtering surgery, due to their greater propensity for inflammation and post-surgical scarring. They may also be affected by asymmetric visual acuity from amblyopia, organic causes, binocular visual impairments due to anisometropia, glaucomatous optic neuropathy, and structural changes in the ocular and orbital anatomy and may affect ocular alignment [31].

In cases of heterotropia, the degree of misalignment may exceed the patient's fusion vergence capacity [32]. Our study included patients across various stages of glaucoma, correlating with different severities of visual field loss. Research shows that loss of peripheral fusion from glaucomatous damage can degrade fusional ability, leading to diplopia [33]. This occurs more frequently in patients with mild visual field loss, whereas those with advanced glaucoma and severe visual field defects may not notice diplopia, potentially leading to underreporting of symptoms [34]. Thus, the absence of reported double vision does not necessarily imply unaffected ocular motility.

Some cases of exotropia may also be explained by a "leash" and "reverse leash" effect, where increased tension in the lateral rectus muscle during convergence causes greater exotropia at near distances [35]. This type of convergence insufficiency derived exotropia is primarily mechanical. Although fusion convergence may overcome this imbalance, severe loss of peripheral vision in advanced glaucoma often undermines the fusion mechanism.

Thus, it is essential to conduct a thorough preoperative motility examination and adnexa assessment for patients being considered for filtering surgery, potentially including presurgical photography for documentation. Strabismus surgery can reduce or correct motility disturbances, but it is important to consider the potential impact on the function of the glaucoma filter. This possibility should be discussed with ophthalmologists who specialize in glaucoma surgery to devise an optimal surgical plan that carefully balances risks and benefits. In some cases, prismatic correction may be a viable alternative [9].For ptosis management, a cautious approach is recommended, allowing for a period of observation to assess whether acute postoperative ptosis stabilizes or resolves on its own before deciding on intervention [10].

## Conclusions

Although in our cohort the occurrence in of ptosis (3,7%), strabismus (1,7%) or the combination of both (1,0%) following glaucoma surgery and drainage device implantation was uncommon, it is crucial to inform patients of these potential complications.

## Supporting information

**S1 File. Database glaucoma (1).**
(XLS)

## Author contributions

**Data curation:** Daniele Ferrari, Irene Schiavetti.

**Formal analysis:** Irene Schiavetti.

**Investigation:** Federica Milanesi, Silvia Acerra, Luigi Peci, Arianna Rizzi, Silvio Semeria.

**Software:** Irene Schiavetti.

**Supervision:** Carlo Alberto Cutolo, Michele Iester, Aldo Vagge.

**Writing – review & editing:** Carlo Catti, Federica Tessitore.

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
