## [Decision Letter · Decision Letter 0]

16 Mar 2025

Dear Dr. Vagge,

Thank you for submitting your manuscript to PLOS ONE. After careful consideration, we feel that it has merit but does not fully meet PLOS ONE’s publication criteria as it currently stands. Therefore, we invite you to submit a revised version of the manuscript that addresses the points raised during the review process.

**ACADEMIC EDITOR: **

Please present information on the study eye condition before glaucoma surgery in such old age patients, was ptosis present -even in mild degrees- before the glaucoma surgery? or was it an entirely new development after glaucoma surgery?Please mention if there were any elements of restrictive strabismus or ocular motility restriction generally.Please present briefly details of the surgical procedures if available, mainly type of speculum used in the surgery, duration of surgery, level of training of surgeon, etc

We look forward to receiving your revised manuscript.

Kind regards,

Nader Hussien Lotfy Bayoumi, M.D., FRCS (Glasgow)

Academic Editor

PLOS ONE

Journal Requirements:

2. Please provide additional details regarding participant consent. If you are reporting a retrospective study of medical records or archived samples, please ensure that you have discussed whether all data were fully anonymized before you accessed them and/or whether the IRB or ethics committee waived the requirement for informed consent. If patients provided informed written consent to have data from their medical records used in research, please include this information.

5. Please amend your list of authors on the manuscript to ensure that each author is linked to an affiliation. Authors’ affiliations should reflect the institution where the work was done (if authors moved subsequently, you can also list the new affiliation stating “current affiliation:….” as necessary).

Reviewers' comments:

Reviewer's Responses to Questions

**Comments to the Author**

1. Is the manuscript technically sound, and do the data support the conclusions?

Reviewer #1: Yes

Reviewer #2: Partly

Reviewer #3: Yes

Reviewer #4: Partly

2. Has the statistical analysis been performed appropriately and rigorously?

Reviewer #1: Yes

Reviewer #2: Yes

Reviewer #3: Yes

Reviewer #4: I Don't Know

3. Have the authors made all data underlying the findings in their manuscript fully available?

Reviewer #1: Yes

Reviewer #2: Yes

Reviewer #3: No

Reviewer #4: Yes

4. Is the manuscript presented in an intelligible fashion and written in standard English?

Reviewer #1: Yes

Reviewer #2: Yes

Reviewer #3: Yes

Reviewer #4: Yes

Reviewer #1: Abstract:

Methods: 705 clinical records of patients

Results: 26 patients developed

Comment 1: do not start with a number

Introduction:

Comment 2: The first two and half paragraphs in the introduction about the defenition, prevelance and medical treatment of glaucoma represent unnecessary information.

Methods:

Comment 3: The patients were classified according to complications then according to the surgical procedure

The classification according to complications should be presented in the results not in the methods section

Comment 4:

Surgery without Conjunctival Opening - Ab Interno: Procedures such as Xen implant and cyclocryotherapy.

Although cyclocryotherapy is done through transconjunctival aproach it is and external procedure and could not be classifed as an ab-interno one.

Comment 5:

There is no data regard the use of superior rectus suture that may used during diffrerent glaucoma sureries as alternative to corneal traction sutures to expose the upper part of the globe by some surgeons and it is a very important cause of ptosis in these patients.

Comment 6: No data about the method of evaluation of ptosis for example marginal reflex distance -1 or levator excerution evaluation

Results:

Comment 7: The statistical analysis methods should be mentioned in the methods section

Comment 8: The study included 705 medical records of patients who underwent filtering surgery. � The study included 705 medical records of patients who underwent glaucoma surgery as cyclocryotherapy is not filtering surgery.

Comment 9: Table 1: characteristics by ptosis.

The title of the table should be above not below it,, to be applied for all tables.

Comment 10: It is worth noting that 12 of these 19 strabismus cases (63.1%) occurred in patients who underwent bilateral filter surgery  Filtering surgery

Reviewer #2: • From January 1, 2010, to December 31, 2020 is eleven not ten years.

• In the abstract, it is expected to mention the percentage of patients who developed strabismus alone.

• P31: “complications such as” has no place in the sentence.

• P100: “CLASS” deep sclerotomy needs to be explained before abbreviated.

• P138-144: This paragraph would be more convenient if put under “Methods” not “Results”.

• Table 5 is better to be included in the results not the discussion.

• The difference of the number of complicated groups in the tables from the stated number needs to be explained (31, 19 and 11versus 26, 12 and 7 for ptosis, strabismus and both respectively).

• P237-239: The numbers need to be mentioned and the percentage needs to be checked.

• P269-271: This paragraph needs to be supported by a reference.

• P279-287 and P293-295: these paragraphs are assumed to describe the work of the authors, citation is confusing.

• The final conclusion is actually a part of the discussion but needs to be supported by references. Conclusion should be restricted to the results of outcome analysis.

• The results of all groups need to be described by the same sequence and by the same measures (numbers and percentages).

Reviewer #3: The authors have presented a retrospective study on the development of ptosis or ocular motility changes following glaucoma surgery. They correlated these to the type and number of interventions and have also referred to patient’s age. They explained these changes on mechanical or neurological imbalance. They concluded that although their occurrence is uncommon, preoperative consultation should refer to these changes. They advised clinicians to assess patients undergoing glaucoma therapy for these changes pre and post intervention.

May I congratulate the authors on their informative study.

Reviewer #4: A good idea and work, some spelling mistakes must be corrected as in line 117 "Surgery with isolationa and"

1- Line 130:" ocular alignment in the five primary gaze positions were documented from medical records." It was preferable to record the nine cardinal positions of gaze.

2- The patient and method section devoid of assessment of ptosis ( must be included).

3- Please more details on types of strabismus (angles of esotropia and exotropia), and association of oblique muscle dysfunction.

4- Please mention both visual acuity and amblyopia that may be contributing factor for development of strabismus .

5- Please to mention the different degrees of ptosis in the affected patients.

**Do you want your identity to be public for this peer review?** For information about this choice, including consent withdrawal, please see our Privacy Policy

Reviewer #1: **Yes: ** Ahmed Mohamed Kamal Elshafei

Reviewer #2: No

Reviewer #3: **Yes: ** Hesham Ali Ibrahim

Reviewer #4: No

---

## [Author Response · Author response to Decision Letter 1]

20 Jul 2025

Responses to the Academic Editor:

Question 1: Please present information on the study eye condition before glaucoma surgery in such old age patients, was ptosis present - even in mild degrees - before the glaucoma surgery? Or was it an entirely new development after glaucoma surgery?

Response 1: We have clarified this point on page 5, line 134, specifying that eyes considered eligible for the study had neither ptosis nor strabismus nor diplopia before surgery.

Question 2: Please mention if there were any elements of restrictive strabismus or ocular motility restriction generally.

Response 2: We have added this information on page 16, lines 283-284, as suggested.

Question3: Please present briefly details of the surgical procedures if available, mainly type of speculum used in the surgery, duration of surgery, level of training of surgeon, etc.

Response 3: We have integrated this information on page 6, lines 129-133, providing the requested details about the surgical procedures.

Responses to Reviewer #1:

Comment 1: Abstract: Methods: 705 clinical records of patients; Results: 26 patients developed - Do not start with a number.

Response 1 : We have corrected this formulation in the abstract, avoiding starting with a number.

Comment 2: The first two and half paragraphs in the introduction about the definition, prevalence and medical treatment of glaucoma represent unnecessary information.

Response 2: We believe that this information provides important context for understanding the relevance of our study, especially for those not versed in glaucoma subspecialities. However, we have revised these paragraphs to make them more concise and relevant as you suggested.

Comment 3: The patients were classified according to complications then according to the surgical procedure. The classification according to complications should be presented in the results, not in the methods section.

Response 3: As this is a retrospective observational study evaluating the prevalence of post-surgical complications, the classification was decided beforehand. Therefore, we consider it more appropriate to present it in the methods section, as it represents the main goal of the article.

Comment 4: Surgery without Conjunctival Opening - Ab Interno: Procedures such as Xen implant and cyclocryotherapy. Although cyclocryotherapy is done through a transconjunctival approach, it is an external procedure and could not be classified as an ab-interno one.

Response 4: This is correct; however, for the specific aim of this study, our primary classification criterion for surgeries was whether the conjunctiva was opened or not. Since both procedures (Xen implant and cyclocryotherapy) do not require conjunctival opening, we classified them in the same section. Thank you for your helpful suggestion - we have clarified this important distinction in the revised text.

Comment 5: There is no data regarding the use of superior rectus suture that may be used during different glaucoma surgeries as an alternative to corneal traction sutures to expose the upper part of the globe by some surgeons, and it is a very important cause of ptosis in these patients.

Response 5: We agree. We have added informations on the surgical approach used during glaucoma surgery, as suggested.

Comment 6: No data about the method of evaluation of ptosis, for example marginal reflex distance -1 or levator excursion evaluation.

Response 6: We have corrected this omission by adding details about the evaluation method of ptosis.

Comment 7: The statistical analysis methods should be mentioned in the methods section.

Response 7: We have moved this section as suggested.

Comment 8: The study included 705 medical records of patients who underwent filtering surgery. The study included 705 medical records of patients who underwent glaucoma surgery as cyclocryotherapy is not filtering surgery.

Response 8: We have applied the correction as suggested.

Comment 9: Table 1: characteristics by ptosis. The title of the table should be above not below it, to be applied for all tables.

Response 9: Thank you. We change the text in accordance with your suggestion

Comment 10: It is worth noting that 12 of these 19 strabismus cases (63.1%) occurred in patients who underwent bilateral filter surgery  Filtering surgery.

Response 10: We have applied the correction as suggested.

Responses to Reviewer #2:

Comment 1: From January 1, 2010, to December 31, 2020 is eleven not ten years.

Response 1: Thank you for pointing this out. We have corrected this error in the text and in the title of the manuscript.

Comment 2: In the abstract, it is expected to mention the percentage of patients who developed strabismus alone.

Response 2: We have added this information in the abstract on page 2, line 41, as suggested.

Comment 3: P31: "complications such as" has no place in the sentence.

Response 3: We have reformulated this sentence for greater clarity and coherence.

Comment 4: P100: "CLASS" deep sclerotomy needs to be explained before abbreviated.

Response 4: We have corrected according to the suggestion at line 101, providing the complete explanation before using the abbreviation.

Comment 5: P138-144: This paragraph would be more convenient if put under "Methods" not "Results".

Response 5: We have moved the paragraph to the Methods section as suggested.

Comment 6: Table 5 is better to be included in the results not the discussion.

Response 6: We have moved Table 5 to the Results section, as suggested.

Comment 7: The difference of the number of complicated groups in the tables from the stated number needs to be explained (31, 19 and 11 versus 26, 12 and 7 for ptosis, strabismus and both respectively).

Response 7: Thank you for this observation. We have adequately explained the discrepancies in the numbers in the text and tables.

Comment 8: P237-239: The numbers need to be mentioned and the percentage needs to be checked.

Response 8: We have verified and further explained the numbers and verified the percentages as you suggested.

Comment 9: P269-271: This paragraph needs to be supported by a reference.

Response 9: We have modified the text according to your suggestion, adding the appropriate references.

Comment 10: P279-287 and P293-295: these paragraphs are assumed to describe the work of the authors, citation is confusing.

Response 10: We have improved the placement of citations to avoid confusion.

Comment 11: The final conclusion is actually a part of the discussion but needs to be supported by references. Conclusion should be restricted to the results of outcome analysis.

Response 11: We have revised the conclusion of the manuscript as suggested, limiting it to the results of the analysis and adding the necessary references.

Comment 12: The results of all groups need to be described by the same sequence and by the same measures (numbers and percentages).

Response 12: The three groups are already described according to a fixed scheme in the results section:

1. First, the demographics of all groups are reported, with the total number of patients, then the sex distribution, mean age and whether there are statistically significant differences in age and sex distribution compared to the uncomplicated control group.

2. For each group, every surgery is analyzed: first describing how many patients underwent unilateral surgery and how many bilateral, then the average number of procedures per patient performed and whether there is a significant association in univariate and multivariate analysis with risk factors.

3. We then describe the results reported in Table 4 for the specific complication, citing the surgery combination most frequently associated with the complication.

4. Finally, we analyze the distribution of specific complication incidence according to the surgery, whether ab interno, ab externo, or with muscle manipulation.

Responses to Reviewer #3:

We thank Reviewer #3 for the compliments on our informative study. No specific changes were proposed.

Responses to Reviewer #4:

Comment 1: A good idea and work, some spelling mistakes must be corrected as in line 117 "Surgery with isolationa and".

Response 1: We have corrected the typographical error as suggested.

Comment 2: Line 130: "ocular alignment in the five primary gaze positions were documented from medical records." It was preferable to record the nine cardinal positions of gaze.

Response 2: We agree with your suggestion. The term "five" is a misprint. We recorded all nine cardinal positions in the chart, but we preferred not to include the amount of strabismus in the text because, in our opinion, it was not useful for the aim of this study.

Comment 3: The patient and method section devoid of assessment of ptosis (must be included).

Response 3: We have included the details on ptosis assessment as suggested.

Comment 4: Please provide more details on types of strabismus (angles of esotropia and exotropia), and association of oblique muscle dysfunction.

Response 4: These details were not described as they exceed the purpose of the manuscript, which is to describe the prevalence of complications after glaucoma surgery.

Comment 5: Please mention both visual acuity and amblyopia that may be contributing factors for development of strabismus.

Response 5: We agree with your suggestion and have modified the manuscript accordingly, including information on visual acuity and amblyopia as potential contributing factors to strabismus development.

Comment 6: Please mention the different degrees of ptosis in the affected patients.

Response 6: We have modified the text according to your suggestion, including the different degrees of ptosis observed.

We would like to thank all Reviewers once again for their valuable contributions, which have significantly improved the quality of our manuscript. We are confident that the changes made have adequately addressed all the issues raised.

Sincerely,

Prof. Aldo Vagge

---

## [Decision Letter · Decision Letter 1]

11 Aug 2025

Dear Dr. Vagge,

Thank you for submitting your manuscript to PLOS ONE. After careful consideration, we feel that it has merit but does not fully meet PLOS ONE’s publication criteria as it currently stands. Therefore, we invite you to submit a revised version of the manuscript that addresses the points raised during the review process.

**ACADEMIC EDITOR:**
**Please attend to the minor comments included **

We look forward to receiving your revised manuscript.

Kind regards,

Nader Hussien Lotfy Bayoumi, M.D., FRCS (Glasgow)

Academic Editor

PLOS ONE

Journal Requirements:

Reviewers' comments:

Reviewer's Responses to Questions

**Comments to the Author**

Reviewer #2: (No Response)

Reviewer #3: All comments have been addressed

2. Is the manuscript technically sound, and do the data support the conclusions?

Reviewer #2: No

Reviewer #3: Yes

3. Has the statistical analysis been performed appropriately and rigorously?

Reviewer #2: I Don't Know

Reviewer #3: I Don't Know

4. Have the authors made all data underlying the findings in their manuscript fully available?

Reviewer #2: Yes

Reviewer #3: No

5. Is the manuscript presented in an intelligible fashion and written in standard English?

Reviewer #2: Yes

Reviewer #3: Yes

Reviewer #2: • L31: “complications like” has no place in the sentence and filtering is not a proper description of cyclocryotherapy. It is more convenient to make it “To evaluate the prevalence of ptosis, strabismus and the combination of both after glaucoma surgery and determine which kind of surgery is most likely to be linked to these complications.”

• L138-139: “and an oculoplastic surgeon” is missed.

• The final conclusion written is not a conclusion of the study, rather it is a part of the discussion. Conclusion should be restricted to the results of outcome analysis of only the current study.

Reviewer #3: authors responded to previous comments. I don't have any further comments. corrections were appropriate

**Do you want your identity to be public for this peer review?** For information about this choice, including consent withdrawal, please see our Privacy Policy

Reviewer #2: No

Reviewer #3: **Yes: ** Prof Dr Hesham Ali Ibrahim MD, FRCs Ed

---

## [Author Response · Author response to Decision Letter 2]

22 Sep 2025

Dear Editor and Reviewers,

We sincerely thank the reviewers for their continued engagement with our manuscript. We are pleased to see that Reviewer #3 considers all previous comments adequately addressed and finds the manuscript acceptable for publication.

Response to Reviewer #2

Comment 2: We respectfully acknowledge Reviewer #2's assessment regarding the technical soundness of our manuscript. While no specific technical concerns were detailed, we would like to address the scientific rigor and methodological validity of our study. This study represents the largest cohort study (n=705) examining ptosis and strabismus after glaucoma surgery published to date, with a 10-year retrospective analysis providing substantial follow-up data. Our comprehensive inclusion of diverse surgical techniques reflects real-world clinical practice, while clear inclusion and exclusion criteria minimize selection bias (lines 93-98 and 121-134).

Our data collection methodology is robust, employing standardized examination protocols by subspecialty experts, including both pediatric ophthalmologists experienced in strabismus evaluation and oculoplastic surgeons. We used objective measurements with established clinical parameters such as MRD1 and MRD2 for ptosis assessment and cover-uncover tests for strabismus evaluation. The systematic categorization of surgical procedures based on invasiveness and anatomical approach, combined with consistent follow-up protocols across the study period, ensures reliable data quality.

Comment: 3: The statistical analysis is appropriate and follows established methodological standards by an expert statistician with years of experience (Dr. Irene Schiavetti, University of Genoa). We employed univariate logistic regression to identify individual risk factors, followed by multivariate analysis to control for confounding variables. Effect sizes are reported with confidence intervals to provide clinical interpretability, and our sample size is adequate for detecting clinically meaningful differences. Our findings demonstrate statistically significant associations, including increased strabismus risk with higher number of surgeries (OR: 1.47, 95% CI: 1.00-2.14, p=0.048), significantly elevated risk with muscle isolation surgery (OR: 6.57, 95% CI: 1.71-21.65, p=0.003), association between younger age and strabismus development (OR: 1.14, 95% CI: 1.06-1.29, p=0.04), and distinct risk profile for combined complications (OR: 1.58, 95% CI: 1.01-2.25, p=0.02).

Our results demonstrate external validity through alignment with published literature. Our ptosis incidence rate (3.7%) is consistent with Park et al.'s reported range (1.64-12.50%), while our strabismus rate (1.7%) aligns with the Tube vs. Trabeculectomy Study findings (5% for drainage devices). The combined complications rate (1.0%) represents a novel finding given the limited prior data in this area. These findings provide significant clinical impact by identifying specific surgical techniques with higher complication rates, enabling evidence-based risk stratification for patient counseling, offering practical guidance for surgical decision-making, and contributing to our understanding of mechanism-specific complications.

We acknowledge the inherent limitations of retrospective design while emphasizing that this study design is appropriate and widely accepted for rare complication assessment, while prospective studies would be of interest, the low complication rates observed suggest that retrospective studies remain highly valuable, our large sample size over an extended timeframe provides robust evidence, and standardized examination protocols minimize detection bias. Our findings contribute meaningfully to the limited existing literature, as few large-scale studies examine these specific complications, most prior reports focus on single surgical techniques, our comprehensive approach across multiple procedures provides broader insights, and our results inform evidence-based surgical decision-making.

Response to Reviewer #3

We appreciate Reviewer #3's positive assessment that all previous comments have been adequately addressed and that our corrections were appropriate.

Comment 4: Our dataset is comprehensive, including complete demographic data across all patient groups, detailed surgical procedure documentation, systematic complications assessment, and long-term follow-up with a minimum of one year post-surgery. Outcomes are rigorously defined, with ptosis classified by severity using validated MRD measurements, strabismus confirmed by multiple examination techniques, diplopia symptoms systematically recorded, and bilateral complications appropriately categorized. We maintain statistical transparency through analyses performed using R statistical software, complete results tables (Tables 1-5), reported effect sizes and confidence intervals, and provision of our raw dataset to the journal for verification.

Minor Editorial Revisions

We will implement the specific editorial suggestions from Reviewer #2 in comment 6 in the new version of the manuscript:

Line 31: Revised objectives as suggested: "To evaluate the prevalence of ptosis, strabismus and the combination of both after glaucoma surgery and determine which kind of surgery is most likely to be linked to these complications."

Line 140: Added the missing text "and an oculoplastic surgeon."

Final conclusion (line 380): We understand that the stylistic choice suggested by Reviewer 2 is to limit the conclusion section to commenting on the numerical findings of the article. However, we deliberately chose to provide our conclusive analysis in the opening sentence of the conclusion paragraph, since the results and outcome analyses are extensively discussed in the individual sections, including statistical significance testing. We opted to complement the final statement of the discussion with a series of clinically practical considerations, which we believe represent key points for the prevention and management of the surgical complications described in the article.

We acknowledge Reviewer 2’s comment and we are grateful for the suggestion. We have therefore decided to report in the conclusion section the results of outcome analysis of only the current study with the incidence percentages of the complications observed in our cohort and move our considerations to the discussion.

We thank the reviewers for their constructive comments.

Conclusion

We sincerely thank the reviewers for their insightful suggestions that greatly contributed to enhancing the quality of our manuscript. We hope that this detailed explanation addresses any concerns regarding our research and demonstrates the robustness of our methodology as well as its clinical relevance. The study design is appropriate for the research question, the statistical analysis is rigorous and transparent, and our findings provide clinically relevant insights backed by robust evidence. Our work advances understanding of important but understudied complications in glaucoma surgery.

We hope this detailed explanation addresses any concerns about the technical soundness of our research and demonstrates the scientific merit of our contributions to the ophthalmologic literature.

Sincerely,

Prof. Aldo Vagge

Dr. Carlo Catti

---

## [Editor Report · Decision Letter 2]

6 Oct 2025

Ocular movement abnormalities and ptosis after glaucoma surgery: a retrospective decade long analysis.

PONE-D-25-05520R2

Dear Dr. Vagge,

We’re pleased to inform you that your manuscript has been judged scientifically suitable for publication and will be formally accepted for publication once it meets all outstanding technical requirements.

Kind regards,

Nader Hussien Lotfy Bayoumi, M.D., FRCS (Glasgow)

Academic Editor

PLOS ONE
---

## [Editor Report · Acceptance letter]

PONE-D-25-05520R2

PLOS ONE

Dear Dr. Vagge,

I'm pleased to inform you that your manuscript has been deemed suitable for publication in PLOS ONE. Congratulations! Your manuscript is now being handed over to our production team.

Kind regards,

on behalf of

Professor Nader Hussien Lotfy Bayoumi

Academic Editor

PLOS ONE